# Enhanced cadmium binding ability in response to novel modifications in a *Paramecium* cadmium metallothionein *PMCd1*

Hira Nizam[1], Asmara Imtiaz[1], Fareeda Tasneem[2], Farah Rauf Shakoori[2], Soumble Zulfiqar[1], Amina Younas[1], Sidra Mustafa[1], Asra Ghaus[2], Ayesha Zafar[2], Arshia Nazir[1], Muhammad Sajjad[1], Abdul Rauf Shakoori[1,2]*

1 School of Biological Sciences, University of the Punjab, Quaid-I-Azam Campus, Lahore, Pakistan,
2 Institute of Zoology, University of the Punjab, Quaid-I-Azam Campus, Lahore, Pakistan

* arshaksbs@yahoo.com, arshakoori.sbs@pu.edu.pk

## Abstract

Metallothioneins (MTs) are low molecular weight cysteine rich proteins involved in detoxification of heavy metals. They are synthesized in response to metal exposure and can bind to various metals, thus reducing their toxicity and providing protection against oxidative stress. MTs are considered to be efficient bioremediators of heavy metal contaminated industrial wastewater. The present study was aimed at further enhancing the metal binding capacity of a cadmium metallothionein protein PMCd1, reported some time back from this laboratory in protozoan ciliate *Paramecium*, to equip them with more efficient system to deal with metal contaminated water bodies. Three additional cysteine residues were introduced at three different places of the protein by site directed mutagenesis, viz. S20C, R180C and Y185C. The wild type and each mutant of PMCd1 were expressed in *E. coli* BL21 cells. Metal uptake ability of each transformant was determined in the presence of 1 and 2mM $Cd^{2+}$ in the medium. The three mutants showed enhanced metal uptake compared to the wild PMCd1 which underscored the role of additional cysteines in enhanced metal binding ability. Amongst the mutants, the genetically modified organism with S20C mutation exhibited 9.1 fold more metal uptake compared to the control ciliate. This mutant has great potential to clean the cadmium- contaminated water.

## Introduction

A tremendous increase in pollution leading to environmental degradation and climate change due to rapid industrialization has been observed in the recent decades across the world. These negative impacts can be partially mitigated by cleaning the pollutant-laden liquids and solid wastes before discharging them in the environment. Bioremediation through heavy metal resistant microbes with high metal removing ability is already in practice. Enhancing heavy metal binding capacity of these

**Data availability statement:** Supporting Information file has been uploaded.

**Funding:** The author(s) received no specific funding for this work.

**Competing interests:** The authors have declared that no competing interests exist.

microbes can better equip these microorganisms to decontaminate the waste water of pollutants.

*Paramecium,* along with other protozoan ciliates that thrive in industrial wastewater containing heavy metal ions such as $Cd^{2+}$, $Zn^{2+}$, $Cu^{2+}$, $Ni^{2+}$, $Hg^{2+}$ and $Pb^{2+}$ develop specific mechanisms to tolerate these toxic metals [1–3]. These mechanisms include chemical modification into less toxic form, sequestration by metallothioneins or efflux [4,5].

Metallothioneins (MTs) are cysteine rich proteins with about 15–30% of cysteine residues that bind metal ions [6–10]. These residues constitute "metal binding motifs" with repeat sequences such as C-X-C, C-X-C-C, C-C-X-C, C-X-X-C and C-C forms where X is an amino acid other than cysteine (C) [11,12]. MTs are ubiquitous, hydrophilic, cytosolic, cysteine-rich, low molecular weight stable proteins that play a pivotal role in metal homeostasis and detoxification of heavy metals [13]. They function by sequestering excess metal ions, thereby preventing their interaction with sensitive cellular components [14]. MTs bind metals such as cadmium, zinc, and copper through metal-thiolate clusters [11,15,16]. This interaction results in the formation of metal-protein complexes that are stored in vacuoles and eventually expelled from the cell in non-toxic form [17–19]. Metal ions can trigger oxidative stress by generating reactive oxygen species (ROS). MTs function as antioxidants, effectively scavenging these harmful radicals at a rate significantly higher than glutathione [18], thus mitigating oxidative damage. Additionally, MTs can also be induced by a wide variety of stimuli or environmental stressors, including temperature shocks (heat or cold), pH fluctuations, hormones, cytokines, starvation, and various chemicals. Consequently, MTs are considered multistress proteins [13,18]

The MT superfamily classification system recognizes ciliate MTs as family 7, further divided into subfamilies 7a (CdMTs) and 7b (CuMTs) [20,21]. These subfamilies are distinguished by their metal induction patterns (Cd/Zn or Cu) and cysteine residue clustering patterns [22]. MTs have a lesser half-life as compared to other proteins [23]. Hence, these have been less studied due to limited availability for structure analysis and purification. Therefore, despite various metal bioaccumulation studies in some ciliate genera such as *Tetrahymena* [24], *Colpoda* sp. [25] and *Paramecium* sp. [26–28], MT sequences have primarily been reported only in some of the *Tetrahymena* species, such as *T. thermophila* [18,29,30], *T. pyriformis* [29], *T. pigmentosa* [31], *T. tropicalis lahorensis* [32], and *T. rostrata* [22] and *Colpoda inflata* named Col-MT1 [33]. PMCd1 is the only reported MT from *Paramecium* sp. while PMCd1 MTs share similarities with *Tetrahymena* CdMTs (TetCdMTs) and vertebrate MTs (VerCdMTs). It also exhibits distinct characteristics. PMCd1and TetCdMTs are intronless [18,34]. PMCd1, like other MTs, is cysteine-rich (27.1%). TetCdMTs (99–191 amino acids) and PMCd1 (203 amino acids) are considerably longer than VerCdMTs (25–82 amino acids). Their hydropathicity values viz.-0.54 for TetCdMTs [19] and -0.61 for PMCd1 indicate a hydrophilic nature compared to -0.15 average values for VerCdMTs [35]. A typical feature of MTs is a low abundance of aromatic amino acids, while VerCdMTs and TetCdMTs either lack or contain a few aromatic residues, and PMCd1 exhibits a comparatively higher content. However, similar to other MTs, PMCd1 also contains a low frequency of histidine residues.

Considering that in VerCdMTs, all Cys residues participate in heavy metal binding, resulting in a stoichiometry of Cd7(-Cys)20 and assuming a similar binding mechanism, Gutierrez et al. [19] calculated the theoretical metal-binding capacity of several TetCdMTs. Some of these findings were supported by experimental data [36–38], which indicated a metal-to-Cys ratio of approximately 1:3 for both VerCdMTs and TetCdMTs. Applying this approach to PMCd1, which contains 55 Cys residues, predicts a theoretical binding capacity of 18 metal ions. However, this theoretical value for PMCd1 has not yet been experimentally validated.

In this study, an attempt has been made to increase the number of Cys residues and consequently enhance the metal binding capacity of a *Paramecium* cadmium metallothionein (PMCd1). Site-directed mutagenesis is a technique by which nucleotide sequence of a DNA fragment can be altered by using synthetic oligonucleotides which ultimately results in the internal mismatch to direct the mutation [39]. It could be deletion, insertion or substitution of one or more nucleotides [40]. In this study, this technique has been used to increase the number of cysteine residues through single nucleotide substitution. The individual substitutions were made at the three different positions of the PMCd1 polypeptide chain. The wild and each mutant PMCd1 were expressed in *E. coli* followed by measurement of metal uptake and stored by each transformant to assess any enhanced metal binding ability of the protein in response to Cys addition. Surfaces coated with such better metal chelators or organisms expressing these proteins can be used for bioremediation as well as biodetection of the metal ions.

## Results

### Potential cadmium binding pockets in PMCd1

A novel cysteine rich cadmium metallothionein (PMCd1) has been reported by Shuja and Shakoori [41]. The proposed metal-lothionein (203 aa) has 17 $CX_2C$ motifs arranged in six repeats $X_mCX_2CX_3CX_2CX_nCX_2C$ (m = 8–15, n = 5–10); each repeat having three $CX_2C$ motifs with the exception of the sixth repeat with two motifs only (Fig 1). The codon optimized form of the gene termed PMCd1syn [42] was used in this study. For simplicity, in this article, PMCd1syn is described as PMCd1.

Fig 2A shows five potential cadmium binding pockets. Three mutations S20C, R180C and Y185C were proposed in $Cd^{2+}$ binding pockets_1, 3 and 4, respectively (Fig 2B). None of these affected the 3D structure of PMCd1 (Fig 2C, 2D). The sixth repeat in R180C mutant also possesses the same pattern of $X_mCX_2CX_3CX_2CX_nCX_2C$ with three $CX_2C$ motifs as in rest of the protein.

**(A)**    MDKVNNNSCTVCPTLCATCSDANNCTSCDIGYYLDKTNPSAVTCTKCNNPCYGCV
DNATKCTACDQGLVLDSVNHTCNQCSPECTSCDQADPKNCQTCANGYYYNDNN
QCKQCSNLCKTCQDQNGKGENYCTSCFSGFYQPTGQNTCKICKQPCKTCETAEDH
CLTCYDGNFWDSTNFLRKQCQYPCVFCNDLTTCDTCECCK

**(B)**

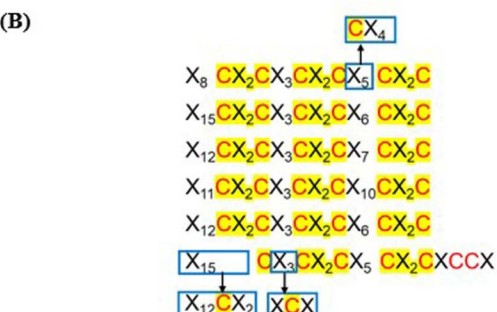

**Fig 1. Wild PMCd1 protein.** (A) Amino acid sequence of the protein. (B) The sequence arranged in repeats $X_mCX_2CX_3CX_2CX_nCX_2C$ (m = 8-15, n = 5-10). Blue boxes show the positions where an X amino acid is replaced by cysteine.

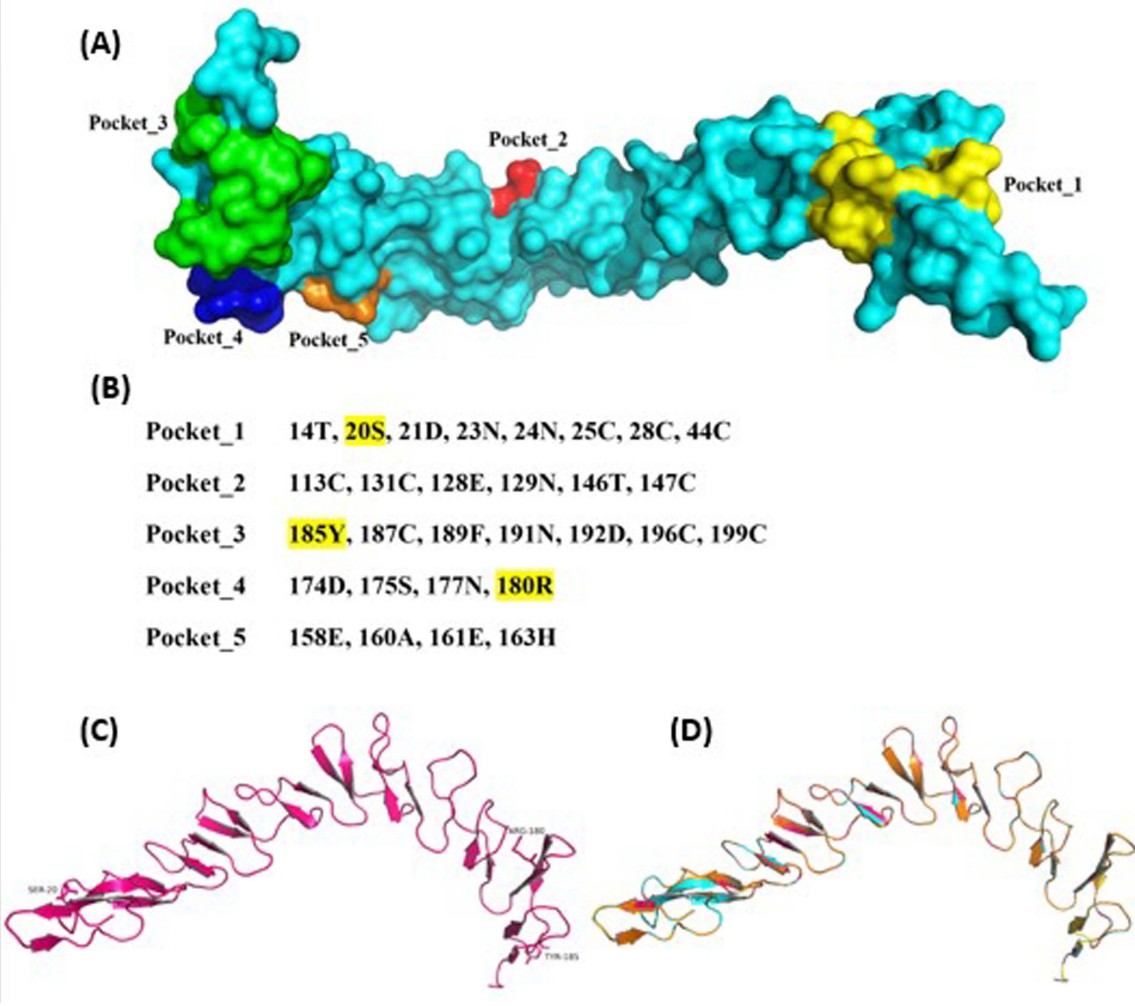

**Fig 2. Prediction of potential Cd² ⁺-binding pockets in PMCd1.** (A) Surface model of metallothionein with its potential Cd²⁺-binding pockets. (B) Sequences of potential Cd²⁺-binding pockets. (C) 3D structure of wild metallothionein protein. Ser20, Arg180 and Tyr185 are represented as sticks. (D) Superimposition of wild metallothionein and its mutants (S20C-cyan, R180C-orange and Y185C-yellow).

## PMCd1 wild and mutant proteins

Fig 3 shows expression profile of whole cell lysates of uninduced and IPTG-induced wild type PMCd1 and its mutant proteins. Protein band at 22.6 kDa represents metallothioneins which are mainly expressed in soluble form as evident when soluble and insoluble fractions of each cell lysate were run on 15% SDS-PAGE (Fig 3).

Before transforming DH5α with recombinant vectors (pET21α-PMCd1, pET21α-S20C, pET21α-R180C and pET21α-Y185C), it was made sure that the recombinant plasmid carried the right sized gene (Fig 4). Fig 4A shows pET21a plasmids containing *PMCd1* and its mutants. Fig 4B shows restriction analysis with *Hin*d III and *Nde* I. Appearance of 612 bp amplicons in Colony PCR confirmed the presence of insert (Fig 4C).

## Effect of Cd on growth of *E. coli* BL21 cells transformed with *PMCd1* mutated genes

Fig 5 shows the effect of Cd⁺⁺ on the growth of *E. coli* BL21 transformants. MIC of BL21 cells against cadmium was determined to be 5mM Culture growth remained unaffected in the presence of 0.5mM Cd²⁺; significantly decreased in the

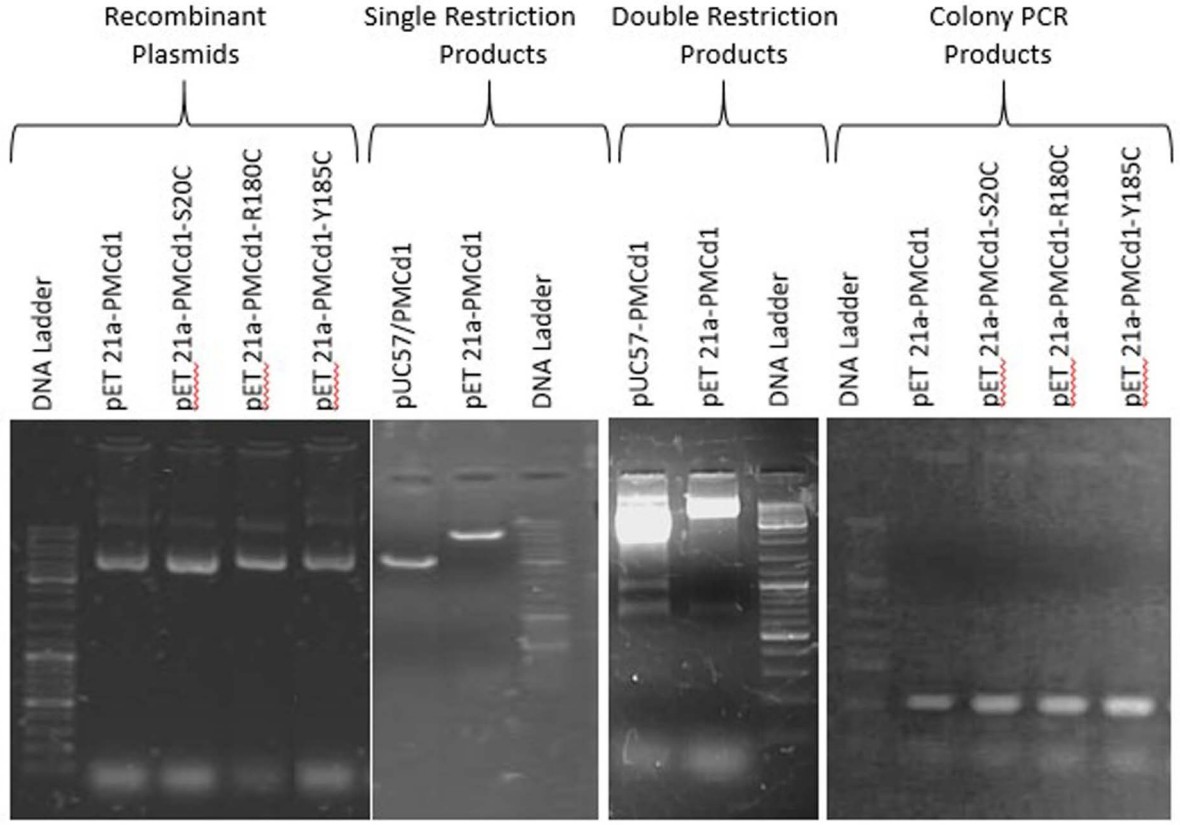

**Fig 3. SDS-PAGE showing expression of recombinant proteins: PMCd1 and its mutants.** (A) Protein profiles of whole cell lysates of uninduced and IPTG induced PMCd1 wild and its three mutant transformants. 22.6KDa bands present only in induced samples are respective metallothioneins. (B) Protein profile of whole cell lysate, soluble proteins and insoluble proteins. Wild and all mutant forms expressed mainly in soluble form. L: Protein ladder (Benchmark cat.no. 10747-012), T: Total proteins in cell lysate, S: Supernatant fractions containing soluble proteins, Is: Pellet fraction containing insoluble proteins.

presence of 1 and 2mM $Cd^{2+}$; and drastically reduced in the presence of 3 and 4mM (Fig 6). Thus $Cd^{2+}$ up to 0.5mM was termed as non-lethal; 0.5-2mM as sub-lethal and 2-4mM as near-lethal.

### Uptake of $Cd^{2+}$ by BL21 cells transformed with wild *PMCd1* gene and its mutants

The dry cell weight of untransformed BL21 cells was calculated to be 0.87mg/ml for OD 1.00. While the wet weight for OD 1.00 was 3.11mg/ml. *E. coli* BL21 cells transformed with pET21- PMCd1 wild showed ability to uptake and store $Cd^{2+}$ in them. The amount of $Cd^{2+}$ stored in the cells increased gradually with the passage of time. All the amounts of cadmium (μg/mg dry cell wt.), described in this section, refer to the amount of the metal bound only with the metallothionein PMCd1 as described in section of materials and methods. Maximum $Cd^{2+}$ bound with wild PMCd1 was found to be 1.20 and 1.16 μg/mg dry cell wt. at 24 h post metal addition in the presence of 1 and 2 mM $Cd^{2+}$ in the medium, respectively (Fig 7A).

Cells carrying mutants containing additional cysteine residue at respective positions showed more uptake as compared to wild PMCd1. R180C showed 3.48μg/mg uptake of $Cd^{2+}$ for 1mM and 2.88μg/mg for 2mM (Fig 7B). This uptake was higher with respect to wild PMCd1 which defines that this mutation is beneficial for MT role. The Y185C mutant showed 2.49μg/mg for 1mM and 2.43μg/mg for 2mM. There was a significant uptake after 6h for 1mM though no significant difference was observed after 24h of incubation as shown in Fig 7C. The mutation S20C resulted in 10.84μg/mg uptake at

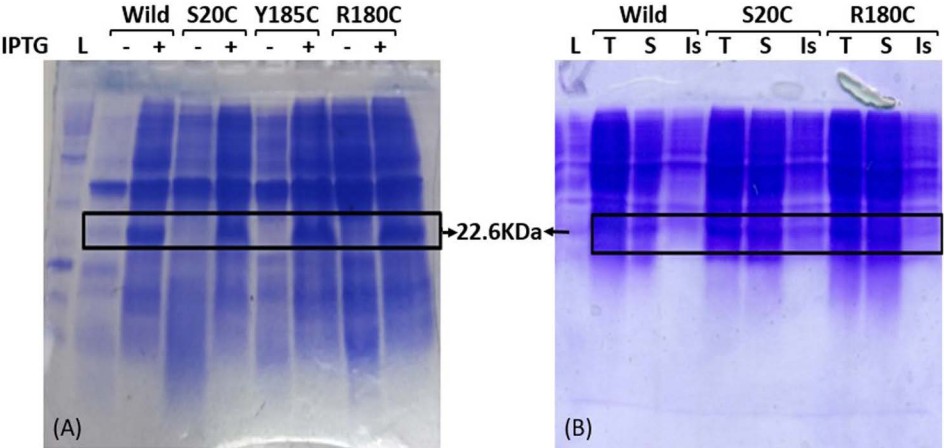

**Fig 4. Recombinant plasmids of *PMCd1* and its mutants.** pET 21a Recombinant plasmids containing wild (PMCd1) and three mutant forms (S20C, R180C and Y185C) were isolated and their integrity was checked on 1% agarose gel. Single restrictions with *Hind* III and double restrictions with *Nde* I and *Hind* III were performed to confirm the presence of right sized insert in cloning vector pUC57 and in expression vector pET21a. Colony PCR was also performed to further confirm the presence of wild *PMCd1* and its three mutant forms in pET21. 1kb DNA ladder (Fermentas SM0331) was used in each gel.

1mM concentration of cadmium and 6.71µg/mg for 2mM (Fig 7D). This uptake was highest as compared to the other two mutations in PMCd1 and proved to be the best mutation for enhancing the cadmium binding efficiency of PMCd1 in *Paramecium*. In all cases, uptake at 2mM was less as compared to 1mM in accordance with its sub-lethal nature.

Fig 8A shows the comparison of cadmium uptake by wild PMCd1 and the three mutants. Fold increase depicted in Fig 8B further highlights the remarkable enhanced binding ability of S20C as compared to the other two mutations.

### Bioinformatics analysis by docking

Docking of $Cd^{2+}$ with the three binding pockets with mutations in PMCd1 metallothionein was performed (Fig 9). The energy (kcal/mol) estimated was -0.1 for each pocket while rmsd values for pockets_1, 3 and 4 were 2.312, 3.808 and 3.528, respectively (Table 1). Moreover, the docking results with AutoDock Vina highlighted that charge of $2^+$ on cadmium was retained in case of pocket_1 while in case of the other two pockets cadmium lost its charge.

### Docking of other cations into the pocket_1 of metallothionein

Docking of cations ($Na^+$, $Ca^{2+}$, $Zn^{2+}$ and $Hg^{2+}$) with atomic radii similar to $Cd^{2+}$ revealed that binding energy (-1.0kcal/mol) and rmsd value (2.312) were same as for cadmium ion (Fig 9). In case of $Cu^{2+}$ with lesser ionic radius different amino acids formed interactions with the metal ion without retaining the charge of 2+ on Cu. Hence, it can be concluded that the pocket_1 of PMCd1 binds to all the metal ions having the ionic radii like $Cd^{2+}$ regardless of charge on the metal ion.

## Discussion

### Why was *Paramecium* chosen as a modal organism

The absence of a cell wall in the vegetative stage of these eukaryotic microorganisms offers a higher sensitivity to environmental pollutants compared to yeasts and bacteria, and hence evince a faster cellular response. Most of the previous work reported from other laboratories on metallothionein in ciliates has been done on *Tetrahymena.* For this study,

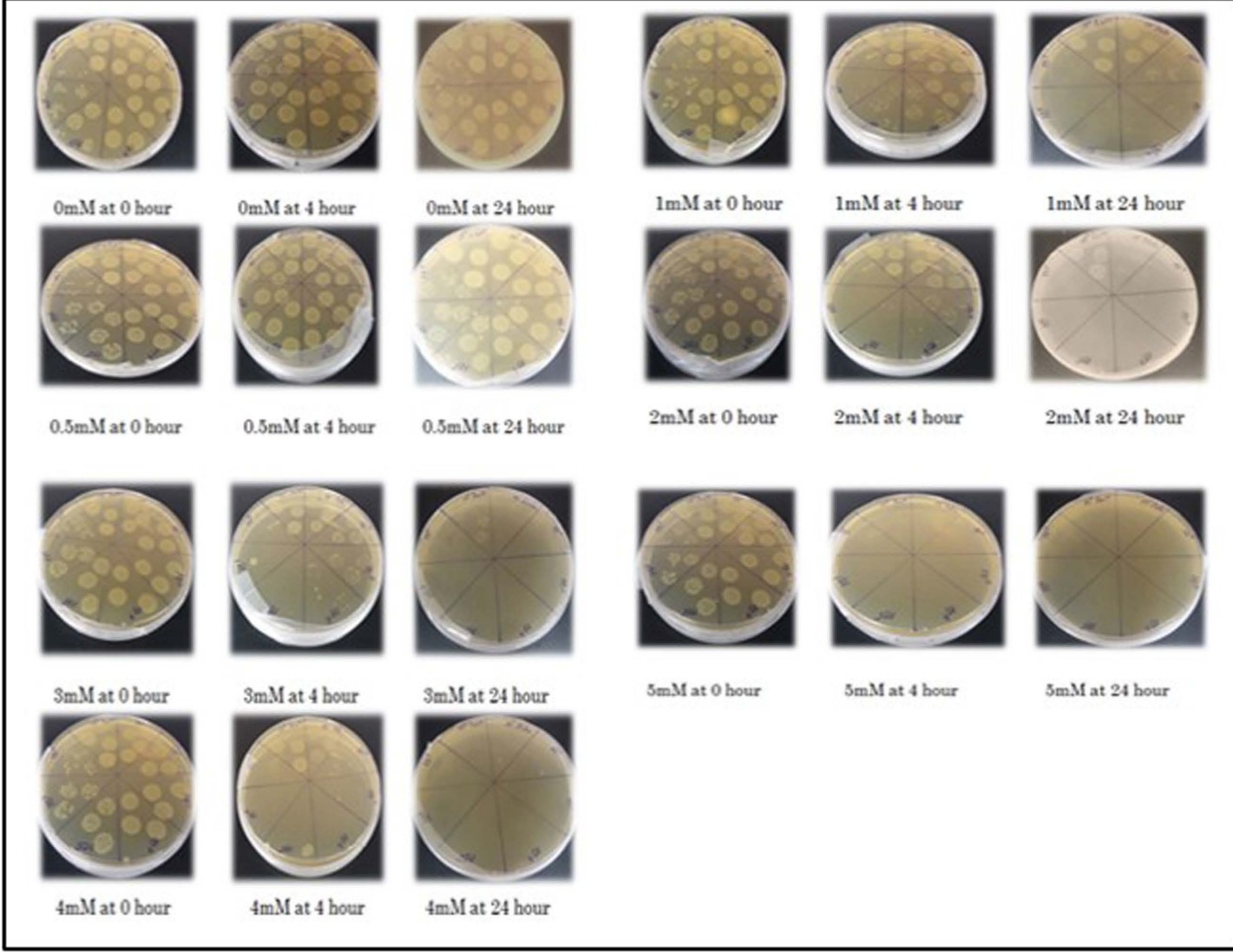

**Fig 5. Effect of different concentrations of cadmium on growth of BL21 cells over a period of 24 h.** Cells grown in the presence of 0-5 mM Cd$^{2+}$ were serially diluted and plated on LB agar plates.

*Paramecium* spp. were selected for their convenient availability in the local environment, and relatively large size (40–50 lm), making their manipulations easier. They grow rapidly giving a high cell density (approximately 10$^6$ cells/ml) in a variety of media and different culture conditions. Moreover the comparison of metallothionein proteins of these two different species of ciliates were expected to provide interesting comparison and new lines of investigation regarding evolutionary development of these microorganisms.

PMCd1, a cysteine rich protein, is the first cadmium metallothionein discovered in *Paramecium* [17]. The objective of this study was to observe the effect of cysteine addition on cadmium binding ability of the protein.

The metal binding capacity of a protein can be enhanced either by introducing more cysteine residues in the protein by site directed mutagenesis or modifying the structure of metallothionein to create more favourable binding sites for metal ions. In this study we achieved the objectives by site directed mutagenesis.

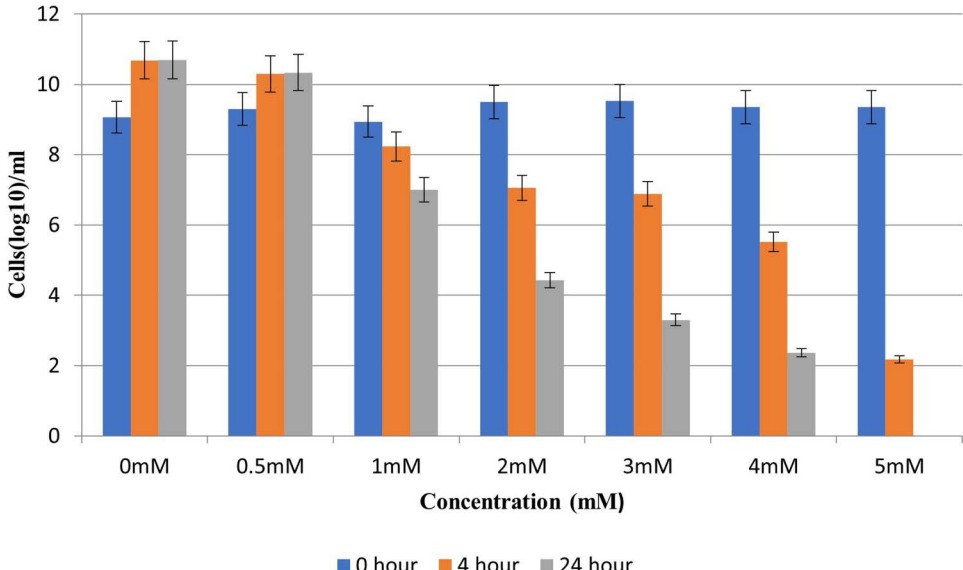

**Fig 6. Effect of different concentrations of cadmium on growth of BL21 cells in the liquid culture medium.** Number of cells (log10/ml culture) was counted and plotted to determine non-lethal, sub-lethal and near-lethal ranges of $Cd^{2+}$ for BL21 cells.

PMCd1 wild and the three mutants were subjected to bioinformatics where possible cadmium binding sites were found by MIB2. The binding sites in close proximity were merged together to form potential binding pockets. This was further confirmed by the predicted 3D surface model of metallothionein. Bioinformatic analysis could not be found in literature for *Paramecium* but one metallothionein from plant *Coptis japonica* has undergone *in-silico* analysis wherein 3D structure of MT has been predicted and after its validation the possible metal binding sites for cadmium ion have been predicted [42].

*E. coli* BL21 cells were independently transformed with wild and mutated *PMCd1* and expression at 22.6kDa was determined by SDS-PAGE in soluble form. In a previous study, same *PMCd1* wild gene was cloned in pET41a vector and expressed with GST tag in *E. coli* BL21 cells [43]. $Cd^{2+}$ binding ability of the protein was determined in the presence of 0.05–0.25 mM metal in the medium. The protein exhibited $Cd^{2+}$ binding ability with a range of $1.4 \times 10^{11}$-$1.95 \times 10^{11}$ $Cd^{2+}$ ions taken up per cell due to *PMCd1* gene. However in this study, wild and mutants of *PMCd1* were expressed without any tag in *E. coli* BL21 cells. The MIC of cadmium against BL21 cells was found to be 5mM. Thus, cells transformed with wild *PMCd1* along with the mutants were exposed to 1 and 2mM $Cd^{2+}$. Wild PMCd1 showed 1.20 and 1.16µgCd$^{2+}$/mg dry cell weight uptake in the presence of 1mM and 2mM $Cd^{2+}$, respectively in the medium.

There was more cadmium uptake by the cells with mutants compared to cells with wild PMCd1. In the presence of 1mM $Cd^{2+}$, the S20C, R180C and Y185C showed uptake of 10.84, 3.48 and 2.49µg $Cd^{2+}$/mg dry cell wt., respectively. This result clearly depicted that the site-directed mutagenesis in wild PMCd1 has increased its cadmium binding capacity as cadmium binds more strongly with the new cysteine motifs formed after mutagenesis and overall, its role as a metallothionein was enhanced. Moreover, the highest uptake ability was observed with S20C where after mutation a C-C motif was formed. The presence of two consecutive Cys might have posed strong binding of cadmium and hence more uptake was seen due to this mutation (Fig 10A). In Y185C, C-X-C motif was formed which might had less strong binding of cadmium (Fig 10C) and therefore less uptake as compared to that of S20C modification. As compared to Y185C, slightly more uptake was determined in R180C where C-X-X-C motif was formed (Fig 10B).

$Cd^{2+}$ docking also showed that pocket_1 was the best amongst the three proposed binding pockets. rmsd value (2.312) was quite low while performing the docking of $Cd^{2+}$ into the first pocket as compared the other two pockets. Moreover, the

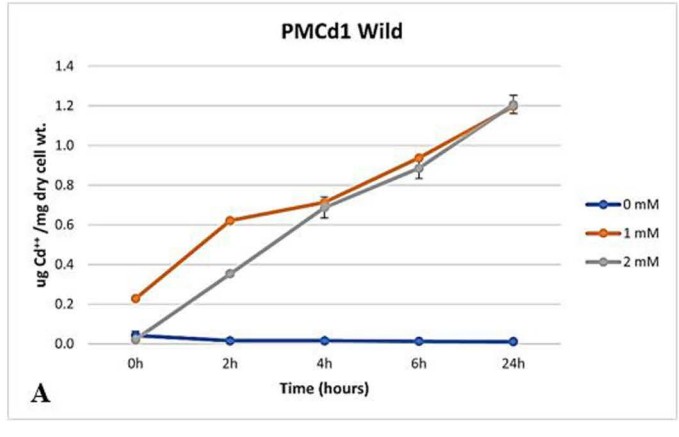
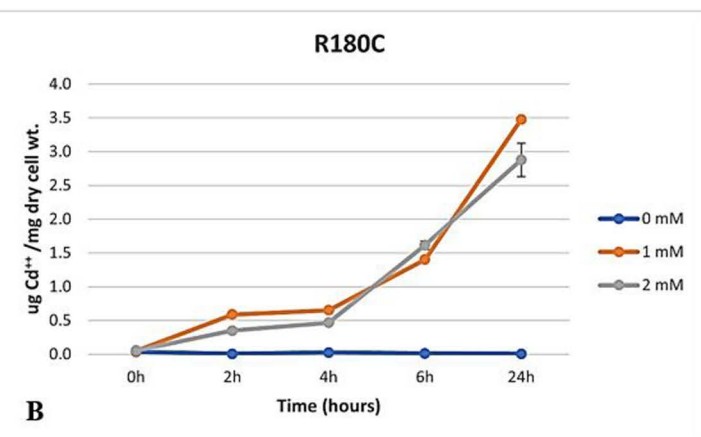
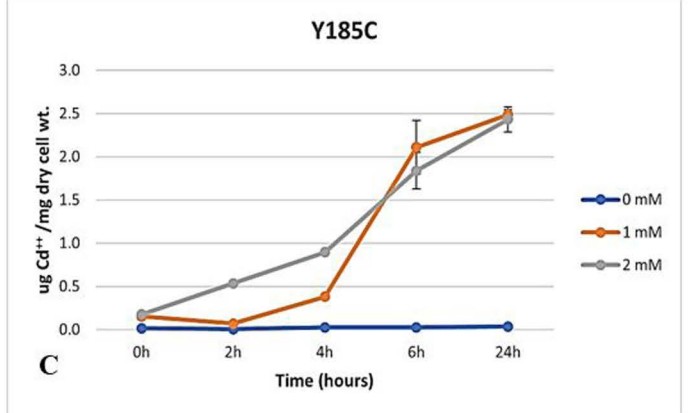
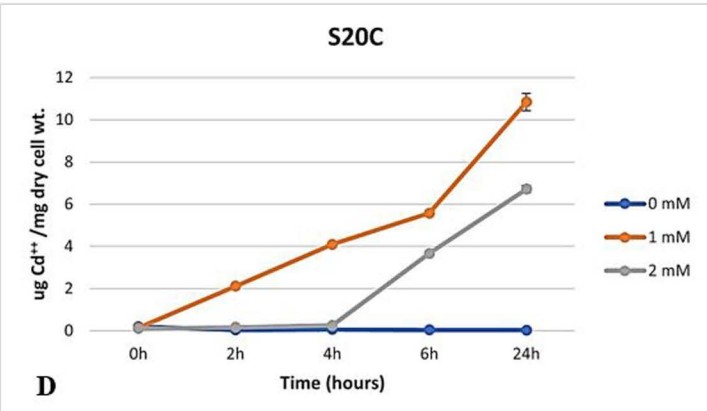

**Fig 7. Cadmium uptake (in µg/mg dry cell weight) by wild PMCd1 and mutants (R180C, Y185C and S20C) over a period of time showing more uptake at 1mM concentration of Cd.**

docking results with AutoDock Vina highlighted that the residues in the pocket_1 were only involved in metal ion contact while retaining the charge of 2+ on cadmium. Hence, based on *in silico* analysis and wet lab work, it can be inferred that pocket_1 is the best possible potential binding site for $Cd^{2+}$.

Docking of other cations into pocket_1 of metallothionein indicated that pocket_1 is a potential binding site for monovalent and divalent cations having same ionic radii as $Cd^{2+}$. Copper with lesser ionic radius could not maintain its charge after docking. No such work has been reported in *Paramecium* species, but some systematic bioinformatics have been reported in another ciliate *Tetrahymena thermophila* where $Zn^{2+}$, $Cu^{2+}$ and $Cd^{2+}$ binding proteins were identified in proteome of *Tetrahymena* sp. through RDGB (Retrieval of Domains and Genome Browsing). The greatest numbers of binding sites were found for $Cd^{2+}$ (2295) which overall helped in the cellular defense against heavy metals and regulation of cell cycle and death [44].

## Mechanism of enhanced cadmium resistance

The enhanced cadmium binding contributes to organism's resistance to cadmium toxicity. This resistance is attributable to (1) production of metallothionein and phytochelatins by the organism which bind Cd ions, sequestering them into vacuoles and thus prevent them from interacting with cellular compounds [17,45,46] (2) transportation of Cd ions out of cells, reducing the intracellular Cd concentration and mitigating its toxic effect [17,47]. (3) enzymatic detoxification of cadmium by

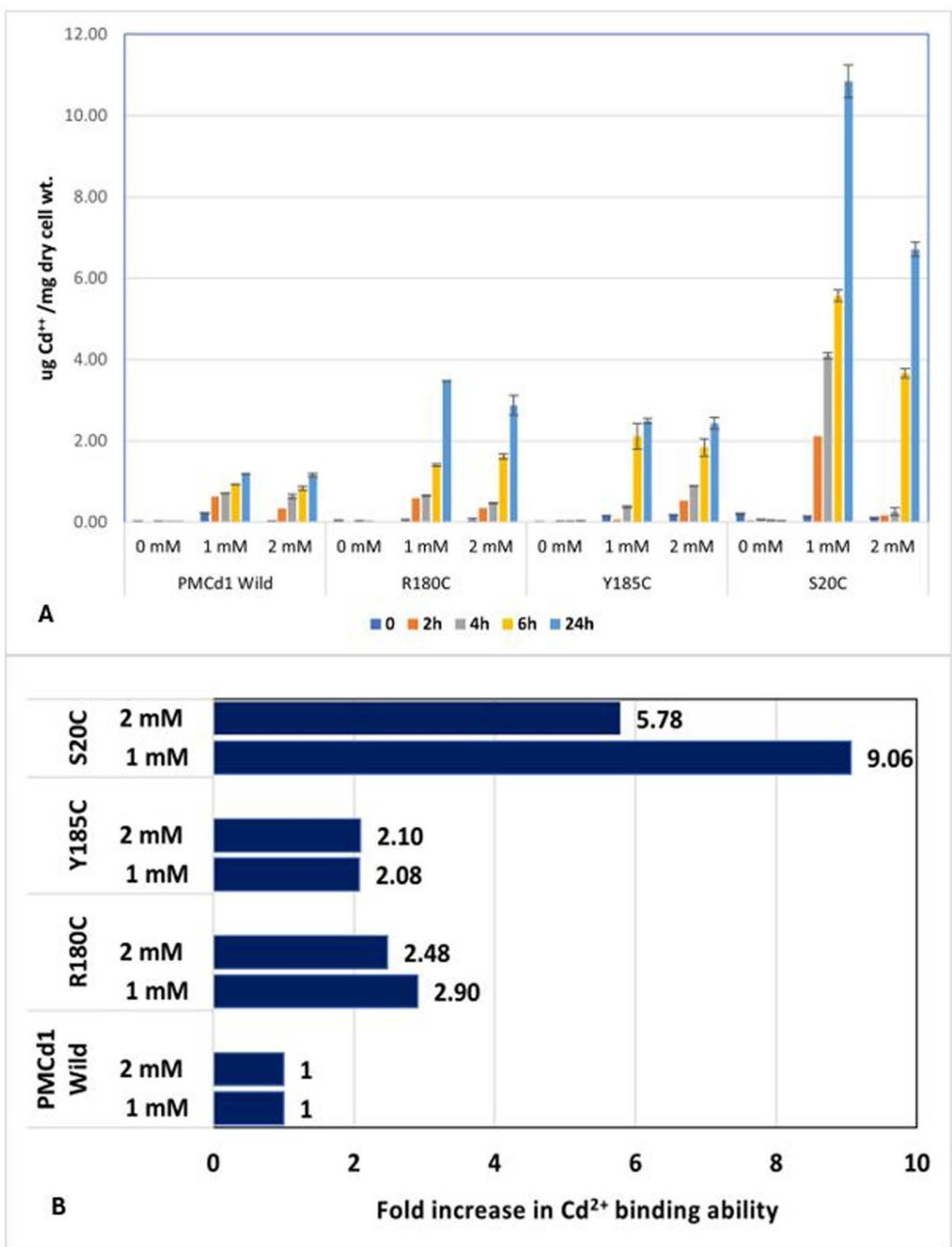

**Fig 8. Comparison of Cd binding capacity of wild type and the three mutants.** (A) Significant differences ($p<0.05$) in uptake as compared to respective control in wild PMCd1 are represented by asterisks. (B) Fold increase in $Cd^{2+}$ binding ability in response to each mutation as compared to wild PMCd1 is given in front of respective bar.

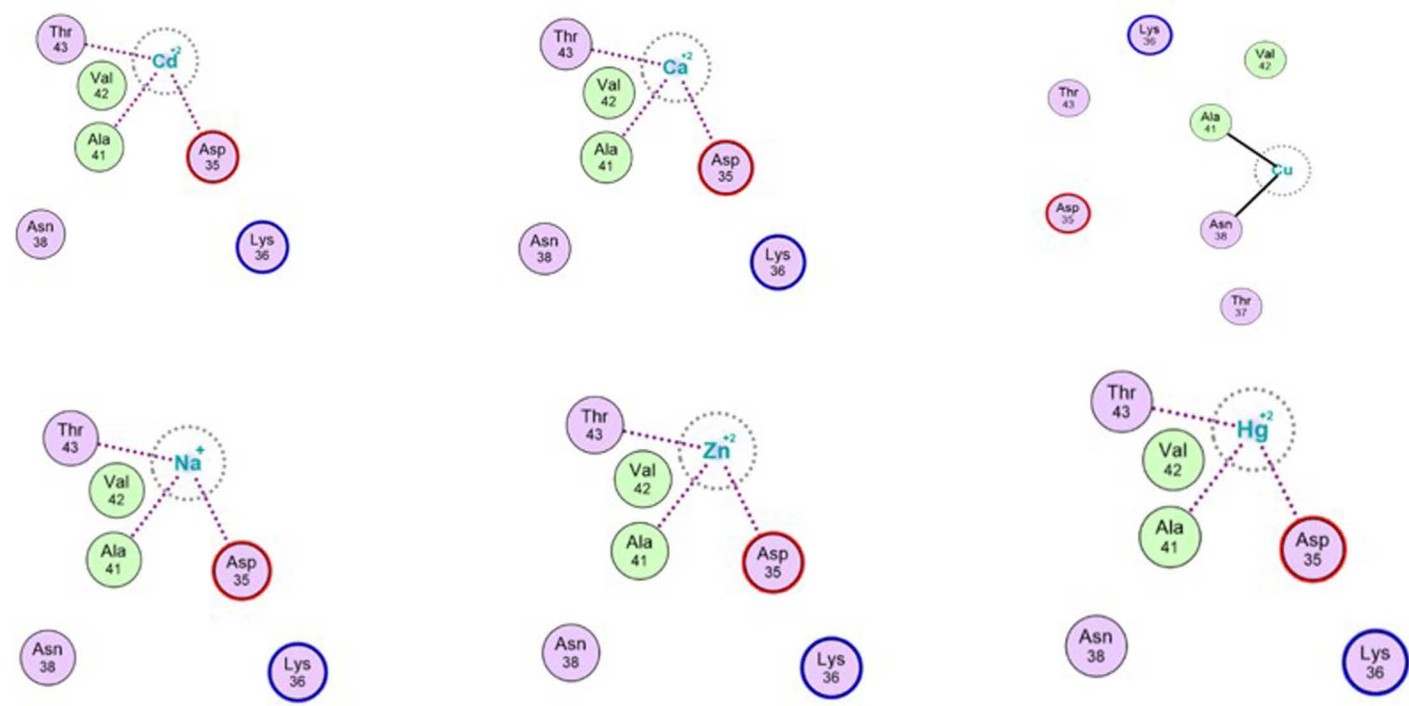

**Fig 9. Docking of Cd² ⁺ into 3 binding pockets of wild type metallothionein. Docking of Cd²⁺ and other cations into the pocket_1 of metallo-thionein.** Like Cd²⁺, in case of monovalent cation Na⁺ and divalent cations Ca²⁺, Hg²⁺ and Zn²⁺, Asp35, Ala41 and Thr43 form direct interactions with the metal ion while Lys36, Asn38 and Val42 are involved in hydrophobic interactions with the metal ion. Amino acids with blue boundaries are alkaline in nature while amino acids with red boundary are acidic. Amino acids represented with green background are hydrophobic in nature. Docking of Cu²⁺ into the pocket_1 of metallothionein showed that Asp35, Ala41 and Thr43 formed direct interactions with the metal ion without retaining the charge of 2+ on Cu. Lys36, Asn38 and Val42 were involved in hydrophobic interactions with the metal ion.

**Table 1. Docking of Cd²⁺ into 3 binding pockets of wild type metallothionein. 2-D interaction are shown in Fig 9.**

| Cd²⁺-binding Pockets | Energy (kcal/mol) | rmsd |
|---|---|---|
| 14T, 20S, 21D, 23N, 24N, 25C, 28C, 44C | -1.0 | 2.312 |
| 185Y, 187C, 189F, 191N, 192D, 196C, 199C | -1.0 | 3.808 |
| 174D, 175S, 177N, 180R | -1.0 | 3.528 |

reducing the availability of free cadmium ions, thereby decreasing ROS production and mitigating oxidation stress. Some enzymes can detoxify Cd by converting it into less harmful forms. Fort example glutathione reductase and other antioxidant enzymes can neutralize the oxidative stress caused by Cd [17,48], (4) development of structural and genetic adaptations that enhance their ability to bind and detoxify cadmium. For example several functional groups (such as amines, carboxyl, phosphate and hydroxyl) on the bacterial cell wall can bind with cadmium ions, thus preventing their entry into the cell [46,47], and (5) enhancement of the expression of genes involved in cadmium binding and detoxification such as Signal molecules like phytohormones and ROS activating pathways. Transcription factors such as WRKY, MYB, bZIP play a key role in regulating these responses [17].

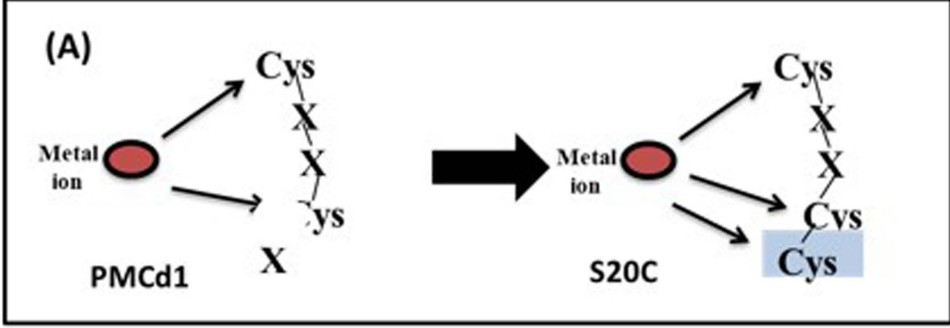

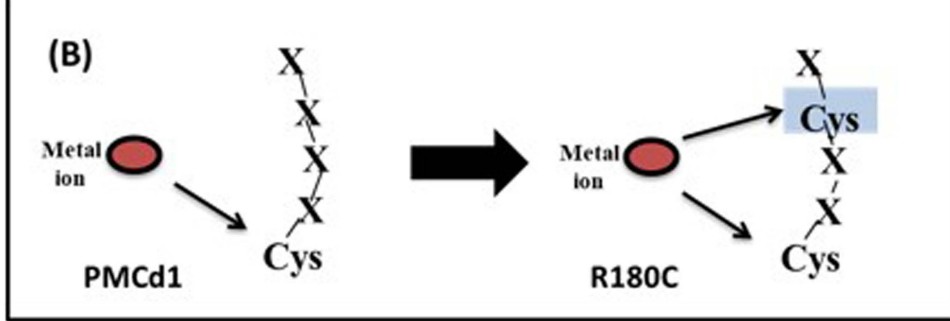

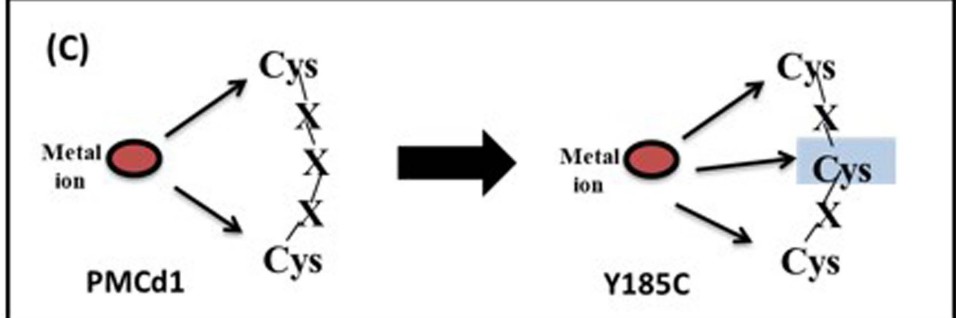

**Fig 10. Structural motifs study after site directed mutagenesis in PMCd1.** Presence of CXXCC motif in S20C **(A)** resulted in maximum enhanced Cd²⁺ binding ability as compared to other two CXXC and CXCXC in R180C **(B)** and Y185C **(C)** mutants, respectively. Additional Cys residue in each mutant is highlighted with blue color.

**Table 2. Primers used for amplification reactions to introduce mutations in *PMCd1*.**

| Mutation | Primers sequences (5'→ 3') | Position |
|---|---|---|
| S20C | GTGCTACATGTTGCGATGCTA TAGCATCGCAACATGTAGCAC | g.58A>T |
| R180C | GCACCAACTTTCTCTGTAAACAATGTC GACATTGTTTACAGAGAAAGTTGGTGC | g.538C>T |
| Y185C | CAATGTCAATGCCCTTGTGTA TACACAAGGGCATTGACATTG | g.554A>G |

\* Bold and underlined nucleotides are different from *PMCd1* gene sequence. These were used to introduce Cys residues. Nucleotides, underlined only, are also different ones resulting in silent mutation. These were used to increase GC ratio of the primer

## Implications of enhanced cadmium binding

Paramecium metallothionein with enhanced cadmium binding capability can serve as bioindicator for cadmium pollution in aquatic environment. Its sensitivity to cadmium makes it a useful organism for monitoring and assessing the level of cadmium contamination in water bodies. Its implications fall within the domain of bioremediation: By harnessing their natural ability to sequester and detoxify cadmium, effective methods to reduce cadmium levels in polluted water bodies and soil can be developed. These methods will incur less financial investment for a more sustainable and long term solution. This technology is likely to have lower environmental impact compared to chemical or physical methods.

## Materials and methods

### Site directed mutagenesis

For site directed mutagenesis, 3D structure of the protein was firstly predicted through SWISS_MODEL and AlphaFold. Possible $Cd^{2+}$ binding sites in the proposed model were predicted through MIB2 (metal ion-binding) tool which is both a metal ion binding site prediction and docking server. The sites with docking score >1.5 were selected. Amongst these, sites present in close proximity were merged together to form five potential cadmium binding pockets. The mutations were proposed in pocket regions keeping in mind that none of these affected the 3D structure of PMCd1. Amongst these, R180C mutation resulted in the same pattern of $X_mCX_2CX_3CX_2CX_nCX_2C$ as in rest of the protein. Multiple alignment of PMCd1 with the homologous sequences in other *Paramecium* spp. revealed that these contain Cys at position 180 as shown in the following figure. In the figure *Paramecium* sp. Represents PMCd1 while rest are homologouse sequences of hypothetical proteins of other species including CAD8205572.1 [*Paramecium pentaurelia*], XP_001428442.1 [*P. tetraurelia*], CAD8199496.1 [*P. octaurelia*] and CAD8116148.1 [*P. sonneborni*].

```
Paramecium sp.    MDKVNNNSCTVCPTLCATCSDANNCTSCDIGYYLDKTNPSAVTCTKCNNPCYGCVDNATK
P.pentaurelia     GYYVNNNSCTVCPTQCAKCSDANNCTSCDIGFFLDSTNPSAVTCTKCTNPCYGCVDNATK
P.tetraurelia     YYVNGSKVCTLCAAQCAKCSDANNCTGCDIGFFLDKTNPNAVTCTKCTNPCYGCVDNATK
P.octaurelia      YYVNGSNACTVCAAQCAKCSDANNCTGCDIGFFLDSTNPNAVTCTKCTNPCYGCVDNATK
P.sonneborni      GYYANNKTCTICPNQCADCSDANNCTACNKGYFLDKTNPNAVTCSNCNNPCYGCFDNATK
                      ..: **:*.   ** ********.*: *::**.***.****::*.****** .*****

Paramecium sp.    CTACDQGLVLDSVNHTCNQCSPECTSCDQADPKNCQTCANGYYYNDNNQCKQCSNLCKTC
P.pentaurelia     CNACDSGLVLDNVNHTCNQCSPECTSCNSADPKNCQTCANGYYYNNNNQCLQCSNLCKTC
P.tetraurelia     CTVCDSGLVLDDVNHTCNQCSPECTSCSQSDPKNCQSCSPGYYYNNNNQCLQCSNLCKTC
P.octaurelia      CTVCDSGLVLDSVNHTCNQCSPQCTSCSQSDPTNCQSCSPGYYYNNNNQCLQCSNLCKTC
P.sonneborni      CSDCDNGLVLDEVNHTCKQCSPECTTCLQNDPKNCQTCADGYYYNNNNQCLQCSNLCKTC
                  *. **.*****.*****:****:**:*  . **.***:*: *****:**** *********

Paramecium sp.    QDQNGKGENYCTSCFSGFYQPTGQNTCKICKQPCKTCET---AEDHCLTCYDGNFWD--S
P.pentaurelia     SNNN---KDYCTSCLSGFYQPTGQNTCKVCTEPCKTCEN---TGDHCLSCYSGNFWD--S
P.tetraurelia     SQND---KNYCTACYDGFYQPTGQNTCKVCIEPCKTCKNS-TSGDQCLSCYSGDFWD--D
P.octaurelia      SNND---KNYCTSCYAGFYQPTGQNTCKVCIEPCKTCKDS-TSGDQCLSCYSGDFWD--E
P.sonneborni      DQNN---KDFCTSCYDGFYQPNGQNTCKVCKQPCLTCKSQQAEGDNCLTCFKGDYLDNST
                  .:::     :::*:**:*   *****.*******:* :** **:     *:**:*:.*:: *

Paramecium sp.    TNFLRKQCQYPCVFCNDLTTCDTCECCK
P.pentaurelia     TNFLCKQCQYPCVNCNDLTTCVTCEKGY
P.tetraurelia     TNFLCKQCQYPCVSCTNLTTCKTCEKGY
P.octaurelia      PNSLCKQCQYPCVSCTNLTTCVTCEKGY
P.sonneborni      TPPSCVQCVYPCVECQDENTCKSCQKGY
                  .        ** **** * :  .** :*:
```

**Y185C** mutation resulted in the generation of a CXCXC, a known motif of *Tetrahymena* MT [19]. **S20C** mutation was randomly selected to generate CXXCC. The introduction of each mutation was confirmed through sequencing of each of the recombinant vectors (pUC57-PMCd1, pUC57-S20C, pUC57-R180C and pUC57-Y185C).

Cysteine residues were introduced at three different places in the PMCd1 through substitutions by site-directed muta-genesis; S20C, R180C and Y185C, one by one. For introducing each mutation, *PMCd1* gene in *pUC57* was subjected to amplification using primers containing desired sequences (Table 2). Each PCR reaction mixture (25 µl) was comprised of 0.25mM dNTPs, 1X Taq buffer, 1.5mM $MgCl_2$, 1µM each primer, 2.5U Taq polymerase and 30ng of template (recombinant plasmid *pUC57*/*PMCD1*). The amplification profile was set in thermocycler (Eppendorf) for 2 min at 95°C followed by 35 cycles of denaturation at 94°C for 1 min, annealing at 59°C (S20C), 64°C (R180C) and 57°C (Y185C) for 1 min and exten-sion at 72°C for 1 min and final extension at 72°C for 7 min. Each amplicon was subjected to *Dpn* I (Thermo Scientific™ cat # ER1701) treatment for the digestion of the parental strand (without mutation). *Dpn* I (1 µl) was added to each PCR product followed by incubation at 37 °C for 90 min.

*E. coli* DH5 α cells were transformed with these recombinant vectors (pUC57-PMCd1, pUC57-S20C, pUC57-R180C and pUC57-Y185C). Each recombinant vector was purified from respective cells and sent for sequencing to Macrogen, Korea to confirm the desired mutation.

### Cloning of *PMCd1* and its mutants in pET21a

Each gene (1 wild and 3 mutants) was sub-cloned in expression vector pET21a using *Nde* I and *Hin*d III restriction sites. Competent cells of *E. coli* DH5α were transformed with these recombinant vectors (pET21a-PMCd1, pET21a-S20C, pET21a-R180C and pET21a-Y185C) and grown at 37°C in LB medium supplemented with 100µg/ml ampicillin.

These recombinant vectors were isolated from respective transformants through alkaline lysis method [49]. The pres-ence of desired insert was confirmed through restriction analysis using enzymes *Nde* I and *Hin*d III in the presence of 1X tango buffer. The reaction was set up on heat block at 37 °C for overnight. Colony PCR was also performed to check the presence of insert using PMCd1 specific forward primer 5' ACTGTGTGTCCAACTCTATGTGC 3' and reverse primer 5' ACAACATTCACAAGTATCACACG 3'. The reaction mixture (25 µl) consisted of 2.5mM dNTPs, 0.5µM each primer, 1X Taq buffer, 2U of Taq polymerase, 2mM $MgCl_2$ and a bacterial colony suspended in sterile water. The PCR conditions were 95°C for 5 min followed by 35 cycles, each of denaturation at 94°C for 45 sec, annealing at 57°C for 35 sec and extension at 72°C for 40 sec and final extension at 72°C for 10 min.

### Expression of *PMCd1* and its mutants

*E. coli* BL21 cells (expression host) were transformed with each recombinant plasmid [50] and then grown in LB medium supplemented with 100µg/ml ampicillin.

A single colony of each positive transformant was inoculated in 10 ml LB ampicillin broth and allowed to grow at 37°C with shaking at 120 rpm overnight. It was sub-cultured in two flasks each containing 20 ml LB ampicillin medium. The flasks were incubated at 37°C with shaking at 120 rpm for 2 h till the $OD_{600}$ reached 0.5–0.8. IPTG (0.1 mM) was added to one of the two cultures of each transformant. These were further grown for another 6 h at 37°C with shaking at 120 rpm after which cells were harvested through centrifugation at 10,400 x g for 20 min. Each cell pellet was resuspended in an appropriate amount of 20 mM Tris-Cl buffer (pH 7.6) keeping the final $OD_{600}$ of culture 10. These samples were sonicated continuously set on ice at a pulse rate of 30 seconds with a rest of 1 min for 32 cycles until clear. An aliquot of each cell lysate (200 µl) was saved and the remaining sample was centrifuged at 13,680 x g at 4°C for 10 min. Pellets were resus-pended in 20mM Tris-Cl (pH 7.6). Each cell lysate, supernatant and pellet sample was run along with protein marker (BenchMark cat.no. 10747–012) on 15% SDS-PAGE.

### Determination of MIC of Cd against *E. coli* BL21

Miles and Misra method [51] was used to determine the minimum inhibitory concentration (MIC) of Cd for which untrans-formed BL21 cells were used. In the 2h old culture, $Cd^{2+}$ (1M $CdCl_2$ stock solution was prepared in deionized water)

was added with final concentration of 0.5, 1, 2, 3, 4 and 5mM. From each culture, serial dilutions with factor of 10 were prepared at 0, 4 and 24hs after $Cd^{2+}$ addition. Each dilution (5 µl) was spotted on LB agar plate in triplicate followed by incubation at 37ºC for overnight. Number of colonies (CFU) were calculated in each spot. From this data, MIC as well as the effect of $Cd^{2+}$ on cell growth was determined.

## Cadmium uptake by *E. coli* transformed with PMCd1 and its mutants

$Cd^{2+}$ uptake ability of *E. coli* transformed with the wild and the three mutant *PMCd1* genes was determined in the presence of 0, 1 and 2mM $Cd^{2+}$. Overnight culture of each transformant was diluted (1:100) in six 100ml flasks each containing 20ml LB broth prepared with deionized water and supplemented with 100µg/ml ampicillin. These six flasks, two for each of the $Cd^{++}$ concentrations were incubated at 37 ºC, 100rpm for 6hs. $Cd^{2+}$ was added with final concentration of 1mM to first group, 2mM to second group while third group served as control (no $Cd^{2+}$). At the same time, to one flask of each group, IPTG was added with final concentration of 0.1mM that led to expression of respective PMCd1s. At 0, 2, 4, 6 and 24hs post $Cd^{2+}$ and IPTG addition, $OD_{600}$ of each culture was noted down. An aliquot (2ml) was taken out and centrifuged at 13,680 x g for 5min to harvest the bacterial pellet. It was washed with 0.89% saline solution and digested with 50 µl $HNO_{3\,(Conc.)}$ at room temperature for overnight. To the digested pellet, deionized water was added up to 1ml. Amount of $Cd^{2+}$ present in the digested pellet ($Cd^{2+}$ uptake) was determined through atomic absorption spectrophotometer (AAS) (Thermo-Unicam-SOLAAR). The instrument parameters included 228.8nm emission wavelength with $0.5_{nm}$ band pass. The flame used was of air-acetylene. Serially diluted standard solutions of $Cd^{2+}$ were used to obtain a normal segmented curve. The amount of $Cd^{2+}$ present per ml in each sample was determined from this linear curve of the standard solutions. µg $Cd^{2+}$ per mg dry cell weight was calculated according to the following mathematical relationship:

$$\mu g\ Cd^{2+}/\ mg\ dry\ cell\ weight = R \times Vs/\ Vc/\ OD_{600}/\ Const.$$

Where R = Reading at AAS; Vs = Volume of sample at the time of reading at AAS; Vc = Volume of culture used to make cell pellet and Const. = Weight of dry cell pellet per ml culture with $OD_{600}$ of the culture 1.00. For determination of Const., 400ml mid log phase culture of *E. coli* BL21 was pelleted down at high speed prior to which its $OD_{600}$ had been measured. The harvested pellet was dried at 65 °C and weighed. Value of Const. was calculated according to the following mathematical relationship: Const. = Weight (mg)/ Volume (ml)/ $OD_{600.}$

   Cadmium found in uninduced transformants (without IPTG) represented the metal uptake ability of *E. coli* itself while cadmium in transformants induced with IPTG represented metal up take by intrinsic ability of *E. coli* as well as due to PMCd1 expressed in the cells. The difference between the two values represented cadmium bound to PMCd1. This value was determined for each of the wild and three mutant forms of PMCd1. The difference between the metal bound to mutated and wild PMCd1 wild represented the enhanced ability of PMCd1 in response to respective mutation.

## 3D structure of the protein

3D structure of the protein was modeled through SWISS_MODEL and AlphaFold. Quality assessment of the model was performed by ProSA and MolProbity [52]. The 3D structure of $Cd^{2+}$ was retrieved from ChemSpider. Both the protein and $Cd^{2+}$ pdb files were converted into pdbqt format after adding polar hydrogens and charges. Using these pdbqt files, possible cadmium ($Cd^{2+}$) binding sites in the proposed metallothionein model were predicted through MIB2 (metal ion-binding) tool which is both a metal ion binding site prediction and docking server [53]. The sites with docking score >1.5 and present in close proximity were merged together.

## Docking of $Cd^{2+}$ and other cations

$Cd^{2+}$ was re-docked into three binding pockets identified earlier by using AutoDock Vina 1.2.x. [52-54]. All source code is available under the Apache License version 2.0 from github.com/ccsb-scripps/AutoDock Vina and the Python package index Pypi.org/project/vina.

The protein-$Cd^{2+}$ complex with the lowest binding energy (kcal/mol) was selected for further analysis by MOE (molecular operating environment) v2022 for 2D interaction analysis.

In order to determine that whether the pocket_1 is specific for $Cd^{2+}$ (ionic radius = 0.95Å) or may serve as binding site for other metal ions, monovalent $Na^+$ (ionic radius 1.02Å) and divalent cations $Ca^{2+}$ (ionic radius 1.1Å), $Hg^{2+}$ (ionic radius 1.02Å) and $Zn^{2+}$ (ionic radius 1Å) having ionic radii similar to $Cd^{2+}$ were docked into the pocket_1 of metallothionein through Autodock Vina. Docking of cation with lesser ionic radius such as $Cu^{2+}$ (0.62Å) into the pocket_1 of the metallothionein was also performed. Autodock Vina has metal ions ($Ba^{2+}$, $Pb^{2+}$ and $Sr^{2+}$) having ionic radii higher than $Cd^{2+}$ couldn't be docked in the pocket_1 of metallothionein due to certain limitations of the software.

## Statistical analysis

Statistical analysis was conducted to compare cadmium uptake by mutants and wild type using Graph Pad Prism software version 9.5.1. Two-way ANOVA was employed to study the experimental data values of atomic absorption spectrometry where the significance level ($P < 0.05$) was determined followed by Tukey's multiple comparisons test.

## Conclusion

Cysteine addition in *Paramecium* cadmium metallothionein PMCd1 by site-directed mutagenesis enhanced the metal binding capacity of the protein Amongst the three mutants, S20C exhibited about 9 and 6 times enhanced Cd binding, than the wild type, in the presence of 1 and 2 mM Cd, respectively, in the medium. In comparison, lesser fold increase in binding by the other two mutants R180C and Y185C (2–3 times more than wild type) was observed. Though the findings of this study substantially prove the enhanced Cd binding capacity of the mutant proteins, however, there is still need to determine stoichiometry metal ions binding per molecule of purified wild and mutant proteins. Moreover, stability assays and structural analyses of the wild and mutant proteins in the absence and presence of Cd may also be helpful for the better understanding of metal coordination with the PMCd1.

## Supporting information

**S1 File.**
(XLSX)

## Author contributions

**Conceptualization:** Abdul Rauf Shakoori.

**Data curation:** Soumble Zulfiqar.

**Funding acquisition:** Abdul Rauf Shakoori.

**Investigation:** Abdul Rauf Shakoori, Hira Nizam, Asmara Imtiaz, Fareeda Tasneem, Farah Rauf Shakoori, Soumble Zulfiqar, Asra Ghaus.

**Methodology:** Abdul Rauf Shakoori, Hira Nizam, Asmara Imtiaz, Fareeda Tasneem, Farah Rauf Shakoori, Amina Younas, Sidra Mustafa, Ayesha Zafar, Arshia Nazir.

**Project administration:** Abdul Rauf Shakoori.

**Resources:** Abdul Rauf Shakoori, Farah Rauf Shakoori.

**Software:** Soumble Zulfiqar, Muhammad Sajjad.

**Supervision:** Abdul Rauf Shakoori, Farah Rauf Shakoori, Soumble Zulfiqar.

**Visualization:** Abdul Rauf Shakoori.

**Writing – original draft:** Abdul Rauf Shakoori, Soumble Zulfiqar.

**Writing – review & editing:** Abdul Rauf Shakoori, Farah Rauf Shakoori, Soumble Zulfiqar.

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
