## [Decision Letter · Decision Letter 0]

6 Jan 2025

PONE-D-24-46664Enhanced cadmium binding ability in response to novel modifications in a Paramecium cadmium metallothionein PMCd1PLOS ONE

Dear Dr. Shakoori,

Thank you for submitting your manuscript to PLOS ONE. After careful consideration, we feel that it has merit but does not fully meet PLOS ONE’s publication criteria as it currently stands. Therefore, we invite you to submit a revised version of the manuscript that addresses the points raised during the review process.

**ACADEMIC EDITOR Comments**Based on the manuscript's quality and the reviewers' suggestions, I inform you that the manuscript cannot be accepted in its current form. A major revision is required to address several critical issues. Key Comments:1. The novelty of the study should be clearly articulated in the Introduction section. This is crucial to highlight the unique contributions of the work.2. The quality of the figures is inadequate, particularly Figures 1 and 6. The authors are requested to provide high-resolution images to ensure clarity and proper visualization.3. Please do not cite irrelevant references as requested by the reviewer.

We look forward to receiving your revised manuscript.

Kind regards,

Veer Singh, Ph.D

Academic Editor

PLOS ONE

Journal Requirements:

3. Please amend your list of authors on the manuscript to ensure that each author is linked to an affiliation. Authors’ affiliations should reflect the institution where the work was done (if authors moved subsequently, you can also list the new affiliation stating “current affiliation:….” as necessary).’

4. We note you have included a table to which you do not refer in the text of your manuscript. Please ensure that you refer to Table I and II in your text; if accepted, production will need this reference to link the reader to the Table.

Additional Editor Comments :

Dear Author,

Based on the manuscript's quality and the reviewers' suggestions, I inform you that the manuscript cannot be accepted in its current form. A major revision is required to address several critical issues.

Key Comments:

The novelty of the study should be clearly mentioned in the Introduction section. This is crucial to highlight the unique contributions of the work.

The quality of the figures is inadequate, particularly Figures 1 and 6. The authors are requested to provide high-resolution images to ensure clarity and proper visualization.

Reviewers' comments:

Reviewer's Responses to Questions

**Comments to the Author**

1. Is the manuscript technically sound, and do the data support the conclusions?

Reviewer #1: Yes

Reviewer #2: Yes

2. Has the statistical analysis been performed appropriately and rigorously? 

Reviewer #1: N/A

Reviewer #2: No

3. Have the authors made all data underlying the findings in their manuscript fully available?

Reviewer #1: Yes

Reviewer #2: Yes

4. Is the manuscript presented in an intelligible fashion and written in standard English?

Reviewer #1: No

Reviewer #2: Yes

5. Review Comments to the Author

Reviewer #1: General Comments:

The study investigates the potential to enhance cadmium binding in a metallothionein (PMCd1) by introducing additional cysteine residues, an approach that is of significant interest in the field of metal detoxification. The findings are promising, particularly the demonstration that the S20C mutation results in a 9.1-fold increase in metal uptake. However, the novelty of this work could be more clearly articulated, especially regarding how the specific placement of cysteine residues enhances cadmium binding compared to existing studies on metallothioneins. The impact of the findings on environmental or biotechnological applications, such as bioremediation, should be more fully explored.

1. A brief discussion of why these sites cysteine residues (S20C, R180C, and Y185C) were targeted and how they relate to cadmium binding in the native protein structure would help clarify the objectives of the study.

2. While the study clearly states that the goal is to enhance metal binding capacity, the hypothesis for why introducing cysteine residues at these specific positions will improve binding should be more thoroughly explained. Are these positions related to known binding sites, or is the enhancement due to structural changes in the protein?

Specific Comments:

Introduction:

1. The introduction should provide a more detailed background on the role of metallothioneins in metal binding and detoxification, particularly in protozoan ciliates like Paramecium. While the significance of metallothioneins in metal homeostasis is mentioned, it would benefit from a more specific discussion of how metallothioneins from ciliates compare to those from other organisms, particularly in their metal uptake abilities.

2. Please explain why Paramecium was chosen as the model organism for this study. How does it compare with other organisms used in similar studies, and why is its cadmium detoxification system particularly relevant for this type of research?

3. In the material and methods section how were the specific mutations (S20C, R180C, Y185C) chosen, and what strategy was used to confirm their introduction into the PMCd1 gene?

4. It would be helpful to include more details on the expression conditions in E. coli BL21 cells. Were any specific conditions used to enhance protein expression or stability? Additionally, were the proteins purified for further analysis, or was the activity assessed directly in the bacterial lysates?

5. The metal uptake assay is a crucial experiment. Please provide more details on the specific methodology used to measure cadmium uptake. For example, were cadmium levels measured using atomic absorption spectroscopy (AAS) or another method? How were the results normalized (e.g., per cell count or protein concentration)?

Results:

The role of the individual cysteine residues (S20C, R180C, Y185C) should be further analyzed. Why does the S20C mutation show the highest increase in metal uptake? Are there any structural or functional reasons behind this observation?

Discussion:

1. The discussion should provide a more in-depth analysis of the mechanisms underlying the enhanced metal binding. For example, how do the introduced cysteine residues affect the protein structure and cadmium binding sites? Do these mutations affect the coordination of cadmium ions or alter the protein’s stability in a way that facilitates greater binding?

2. The implications of these findings for Paramecium’s ability to deal with cadmium contamination should be explored in more detail. How might the enhanced cadmium binding ability contribute to the organism’s resistance to cadmium toxicity, and what are the potential applications of this in environmental cleanup or bioremediation?

Conclusion:

The conclusion should summarize the key findings, emphasizing the practical implications of the enhanced cadmium binding ability. It would be valuable to discuss potential next steps for the research, such as further optimizing the protein for use in cadmium removal from contaminated environments or exploring the stability and effectiveness of the mutant in real-world conditions.

Minor Comments:

1. The manuscript is generally well-written but could benefit from more concise phrasing in certain sections. For example, some sentences in the introduction are lengthy and could be restructured for clarity.

2. Ensure that technical terms, such as "site-directed mutagenesis," are consistently defined and used appropriately throughout the manuscript.

3. The reference list should be expanded to include more recent studies on the engineering of metallothioneins for metal binding enhancement, particularly those focusing on cadmium. Also, ensure that all references are properly formatted according to the journal's style.

4. It may be beneficial to include additional data, such as stability assays or metal binding kinetics, to further support the conclusions drawn from the metal uptake experiments.

Overall Recommendation:

The manuscript presents a valuable contribution to the field of metallothionein engineering for cadmium removal, with promising results for the S20C mutant of PMCd1. However, additional details in the methods, more comprehensive data presentation, and a deeper discussion of the molecular mechanisms underlying the enhanced metal binding are needed to fully strengthen the manuscript. With these revisions, the manuscript has the potential for publication.

Reviewer #2: The manuscript is well written and comprehensive in it's approach but needs further modification for it's consideration for publication in the journal.

Authors, kindly, go through the following comments for needed correction:

1. Revise the discussion section as it appears more of a summary of the work done.

2. Expand the conclusion section.

3. Go through the manuscript for any grammatical error or abbreviations. Use same abbreviations throughout the manuscript, eg, hour is mentioned as h or hs in some sections while hr or hour in figures.

Also the font size is varying in Bioinformatics analysis for docking. Kindly correct it.

4. Mention the versions of the docking softwares used and if possible mention the links of these softwares and sites for accessibility of readers.

5. Explain the rational behind retention of charge on various ions and similar binding ability of metal ions with similar ionic radii as that of cadmium in the discussion section.

6. Significant difference among variables is missing in all the graphs except for Fig 7 (a). Include it in all.

6. PLOS authors have the option to publish the peer review history of their article (what does this mean? ). If published, this will include your full peer review and any attached files.

**Do you want your identity to be public for this peer review?** For information about this choice, including consent withdrawal, please see our Privacy Policy .

Reviewer #1: No

Reviewer #2: No

---

## [Author Response · Author response to Decision Letter 1]

14 Mar 2025

1. the manuscript has been prepared according to PLOS ONE's style requirements, including those for file naming.

2. it is confirmed that all raw data required to replicate the results are included in the manuscript

3. The affiliations of all authors have been correctly mentioned. The Authors FRS, SZ, MS and ARS are permanent employees and have therefore not moved, where as HN , SI , ST, AY , SM , AG , AZ and AN were graduate students in the Institute of Zoology and School of Biological Sciences , University of the Punjab during diffrent periods and have now moved to other parts of Pakistan and abroad in connection with their current profession (unrelated to the work reported in this article ).

4. Table 1 and 2 have now been cited at appropriate places in the text.

---

## [Decision Letter · Decision Letter 1]

24 Apr 2025

Enhanced cadmium binding ability in response to novel modifications in a Paramecium cadmium metallothionein PMCd1

PONE-D-24-46664R1

Dear Dr. Author,

We’re pleased to inform you that your manuscript has been judged scientifically suitable for publication and will be formally accepted for publication once it meets all outstanding technical requirements.

Kind regards,

Veer Singh, Ph.D

Academic Editor

PLOS ONE

Reviewers' comments:

Reviewer's Responses to Questions

**Comments to the Author**

1. If the authors have adequately addressed your comments raised in a previous round of review and you feel that this manuscript is now acceptable for publication, you may indicate that here to bypass the “Comments to the Author” section, enter your conflict of interest statement in the “Confidential to Editor” section, and submit your "Accept" recommendation.

Reviewer #1: All comments have been addressed

Reviewer #2: All comments have been addressed

2. Is the manuscript technically sound, and do the data support the conclusions?

Reviewer #1: Yes

Reviewer #2: Yes

3. Has the statistical analysis been performed appropriately and rigorously? 

Reviewer #1: Yes

Reviewer #2: Yes

4. Have the authors made all data underlying the findings in their manuscript fully available?

Reviewer #1: Yes

Reviewer #2: Yes

5. Is the manuscript presented in an intelligible fashion and written in standard English?

Reviewer #1: Yes

Reviewer #2: Yes

6. Review Comments to the Author

Reviewer #1: The author revised manuscript carefully. Now, the quality of manuscript has been improved as per journal standard. I recommend this manuscript for the publication.

Reviewer #2: (No Response)

7. PLOS authors have the option to publish the peer review history of their article (what does this mean? ). If published, this will include your full peer review and any attached files.

**Do you want your identity to be public for this peer review?** For information about this choice, including consent withdrawal, please see our Privacy Policy .

Reviewer #1: No

Reviewer #2: No

---

## [Editor Report · Acceptance letter]

PONE-D-24-46664R1

PLOS ONE

Dear Dr. Shakoori,

I'm pleased to inform you that your manuscript has been deemed suitable for publication in PLOS ONE. Congratulations! Your manuscript is now being handed over to our production team.

Kind regards,

on behalf of

Dr. Veer Singh

Academic Editor

PLOS ONE